# Efficacy and Safety Assessment of Antifungal Prophylaxis with Posaconazole Using Therapeutic Drug Monitoring in Pediatric Patients with Oncohematological Disorders—A Single-Centre Study

**DOI:** 10.3390/jof11010038

**Published:** 2025-01-06

**Authors:** Karolina Liszka, Paweł Marschollek, Dawid Przystupski, Jowita Frączkiewicz, Monika Mielcarek-Siedziuk, Igor Olejnik, Zuzanna Gamrot, Natalia Haze, Agnieszka Kwella, Paulina Zalewska, Matylda Resztak, Marek Ussowicz, Krzysztof Kałwak

**Affiliations:** 1Department of Pediatric Bone Marrow Transplantation, Oncology and Hematology, Wroclaw Medical University, Borowska 213, 50-556 Wroclaw, Poland; pawel.marschollek@gmail.com (P.M.); dawid.przystupski@umw.edu.pl (D.P.); jowitafr@gmail.com (J.F.); mmielcarek@gmail.com (M.M.-S.); olejnik@olejnik.x.pl (I.O.); zgamrot@gmail.com (Z.G.); hazenatalia@gmail.com (N.H.); akwella90@gmail.com (A.K.); ussowicz@gmail.com (M.U.); krzysztofkalwak@gmail.com (K.K.); 2MonitLab Laboratory, 61-612 Poznan, Poland; paulina@monitlab.pl; 3Department of Physical Pharmacy and Pharmacokinetics, Poznan University of Medical Sciences, Rokietnicka 3, 60-806 Poznań, Poland; mresztak@ump.edu.pl

**Keywords:** posaconazole, therapeutic drug monitoring (TDM), antifungal prophylaxis

## Abstract

Introduction: Posaconazole is recommended for prophylaxis in pediatric immunocompromised patients. Due to its variability in bioavailability and drug-to-drug interactions, EBMT recommends regimens based on therapeutic drug monitoring (TDM). Materials and methods: In total, 171 oncohematological pediatric patients on posaconazole prophylaxis were included. Full pharmacokinetic posaconazole profiles were assessed in 51 children. The efficacy and safety of posaconazole was evaluated by measuring the drug concentration, with dose modification attempted in patients with suboptimal results. The influence of modifying factors on the posaconazole plasma concentration (PPC) was investigated. Results: An insufficient PPC was the main issue, but no significant increase in prophylaxis failure was reported. The modification of the dosage resulted in the optimization of PPC in 50% of patients. No significant correlation between age, gender, diagnosis or the posaconazole dosage and the PPC was found. HCT, total parenteral nutrition and diarrhea were associated with a lower PPC. Hypoalbuminemia was related to both higher and lower PPC. The concomitant administration of specified drugs significantly impacted the PPC. Conclusions: TDM allows the identification of patients receiving non-optimal treatment and offers an opportunity to improve the efficacy and safety of the therapy. However, further research involving larger patient groups and longer observation periods are needed to determine the optimal dosing and target PPC in pediatric patients.

## 1. Introduction

An advancement in the treatment of neoplastic diseases in children results in an increase in overall survival rates. Performing appropriate supportive therapy is no less important. Infections remain one of the main causes of death in oncohematological patients, with fungal infections posing a significant diagnostic and therapeutic challenge [1,2,3]. Appropriate prophylaxis is crucial but data concerning the use of newer antifungal medications in pediatric patients are limited.

Posaconazole, a second-generation triazole, has a broad spectrum of activity against a range of important fungal pathogens, including *Aspergillus* spp., *Candida* spp., *Fusarium* spp. and *Mucoromycetes.* It has a valuable safety profile but shows high intra- and interindividual variability in pharmacokinetics. Its involvement in numerous drug-to-drug interactions (DDIs) determines the final drug exposure, and hence the clinical effect is difficult to predict. Posaconazole is recommended by the European Conference on Infections in Leukemia (ECIL-8) and national committees on primary antifungal prophylaxis in oncohematological pediatric patients [4,5]. Oral suspensions and intravenous solutions are the only available formulations in Poland. Newer pharmaceutical forms offering more stable pharmacokinetic features, such as delayed-release tablets, are still unreachable.

The therapeutic efficiency of posaconazole depends on achieving a steady target plasma concentration. Therefore, regimens should be based on therapeutic drug monitoring (TDM) to ensure treatment efficacy and enhance its safety [4,5]. However, the recommended PPC for primary prophylactic regimens remains undetermined [6,7].

The aim of the study was to evaluate the efficacy and safety of posaconazole antifungal prophylaxis in pediatric oncohematological patients by measuring the plasma drug concentration.

## 2. Patients and Methods

This research was carried out as a single-center prospective study in 2020–2023 at the Department of Pediatric Bone Marrow Transplantation, Oncology and Hematology of Wroclaw Medical University. The protocol for the study was approved by the Ethics Committee of the Medical University of Wroclaw. Informed consent was obtained from all subjects or their parents involved in the study. The trial was divided into two parts, and a total of 171 patients were included. All subjects were administered pozaconazole as a prophylaxis for invasive fungal infection (IFI). The majority of patients received posaconazole oral suspension, but 11 children were administered intravenous solution. Posaconazole oral suspension (40 mg/mL) was administered according to Welzen’s recommendations based on the patients’ body weight, with the following dosages: 10–14 kg: 120 mg; 15–19 kg: 160 mg; 20–24 kg: 200 mg; 25–29 kg: 220 mg; 30–34 kg: 260 mg; 35–39 kg: 280 mg; and ≥40 kg: 300 mg. All doses were administered BID [8]. An intravenous solution was administered as a single daily dose of 150 mg or 300 mg, depending on the patient’s body weight. No loading dose was implemented. Blood samples for the posaconazole plasma concentration (PPC) assessment were taken after a minimum of seven days of therapy, allowing for the establishment of the drug’s steady state. Blood samples were centrifuged (3500× *g* at 20 °C for 10 min) and the plasma was stored at −70 °C until analysis. For each plasma sample, the PPC was determined at the MonitLab^TM^ Laboratory (Poznan, Poland) using validated high-performance liquid chromatography with fluorescence detection (HPLC-FLD). Then, 50 μL of the matrix was used for the analysis. The preparation of the sample for the analysis was based on the one-step precipitation of plasma proteins to eliminate errors associated with the measurement [9]. A PPC of 0.7–3.0 mg/L was considered as the target range [6,8,10,11]. The patients’ characteristics are presented in Table 1.

In the first part of the study, the patients’ posaconazole pharmacokinetic (PK) profiles were analyzed and their PK parameters (C_max_, C_avg_, T_max_, AUC_tf_) were calculated. Their PPCs were assessed in four timepoints—C_0_ (C_trough_)—just before drug administration and then three, five and eight hours after posaconazole intake (C_3_, C_5_, C_8_). The second part of the trial was dedicated to the evaluation of TDM in posaconazole-based prophylactic regimens. The PPC was assessed in one timepoint, namely C_0_, just before posaconazole administration in the case of the oral suspension or just before the start of intravenous infusion. An intervention with posaconazole dose modification was attempted in patients with a PPC out of the therapeutic range. The doses were increased or decreased by 30% according to the PPC. Subsequent measurement of the drug concentration was performed 7 days after modification (allowing for the establishment of the drug steady state).

In both parts of the study, the efficacy (occurrence of breakthrough fungal infection) and safety (incidence of severe drug-related toxicities) of posaconazole were assessed. A one-month observation period after PPC measurement was established. An analysis of the potential influence of selected factors on posaconazole pharmacokinetics and hence final exposure was performed. For each part of the study, a separate statistical analysis was conducted to assess the correlation between the measured PPC and selected variables: age, sex, diagnosis (oncological vs. non-oncological), treatment involving hematopoietic cells transplantation (HCT) or CAR-T cell therapy, the route of nutrition (oral or parenteral), the albumin concentration in plasma, gastrointestinal pathology occurrence (mucositis, diarrhea, vomiting), and medications co-administered with posaconazole. The impact of selected factors on the PK parameters was assessed. Categorical variables were assessed using the chi-squared test. For continuous variables, pairwise correlations were evaluated with Spearman’s rank correlation coefficient. Comparisons involving binary nominal variables were performed using the Mann–Whitney test with continuity correction, accounting for small sample sizes. For nominal variables with more than two categories, statistical significance was determined using the Kruskal–Wallis test. Finally, a logistic regression model was built to predict the probability of untherapeutic PPC occurrence. Statistical analyses were conducted using Statistica 13 software (TIBCO Software Inc. 2017, STATISTICA, version 13, Dell, OK, USA).

A review of potential drug–drug interactions was conducted. Based on the literature analysis, selected drugs were included in the statistical analysis [5,12,13,14]. A detailed evaluation was carried out for the therapeutics of known interaction mechanisms with posaconazole and/or for which suggestive results from the statistical analysis were obtained. Trimethoprim-sulfamethoxazole, used for pneumocystis pneumonia prophylaxis three days a week, all interventional medications and cytostatic drugs were excluded from consideration.

## 3. Results

In the first part of the study, of the 45 patients on posaconazole oral suspension, 22 (49%) reached a posaconazole plasma concentration within the therapeutic range. Twenty patients presented with an insufficient PPC and three patients achieved a concentration above the upper limit. The measurements confirmed that the PPC remains stable in the steady state. The Spearman correlation matrix showed that the concentration measured at timepoint C_0_ significantly correlates with the measurements at the other timepoints: 0.770 for C3, 0.923 for C5 and 0.906 for C8 (*p* < 0.05). The PPC at C_0_ accurately reflects the patient’s overall exposure to posaconazole. The average difference between the highest and lowest concentration in the profile was the highest in patients with a PPC above the therapeutic range—2.66 mg/L; in patients with an insufficient PPC, it was 0.16 mg/L, whereas in patients with the target PPC, it was 0.39 mg/L. Unexpectedly, not always the lowest concentration was detected in C_0_.

In patients administered the intravenous posaconazole formulation, the problem of insufficient drug exposure was not observed. The peak PPC was detected at timepoint C_3_, and the concentration at C_0_ was the lowest, as expected. In two out of six patients, concentrations ≥ 3.0 mg/L were reached in most of the profile assessments. The values of selected posaconazole PK parameters in patients on oral and intravenous posaconazole prophylaxis are presented in Table 2.

Despite the high percentage of insufficient PPCs among the study group, no patient included in the first part of the trial developed a probable or proven IFI during observation. No patient developed side effects related to posaconazole administration. However, in five patients, among those in whom the PPC was ≥3.0 mg/L in at least one measurement, elevated aminotransferases activity was observed without any other clinical or laboratory symptoms of liver function impairment. The maximal aminotransferase activity, at ALAT 337 U/L and ASPAT 167 U/L (normal range ALAT: 0–45 U/L, ASPAT: 0–35 U/L), was observed in a patient with an incidentally high PPC (C_5_). The maximum bilirubin serum concentration noted in patients with an elevated aminotransferase activity was 1.6 mg/dL (normal range: 0.3–1.2 mg/dL).

The significant correlation between the maximum and average concentration (C_max_, C_avg_), as well as the area under the concentration–time curve (AUC_tf_) for posaconazole with the route of administration (oral vs. intravenous), the albumin concentration in plasma (presence or absence of hypoalbuminemia), and co-administration with ondansetron and levetiracetam, was demonstrated as presented in the Table 3. Patients receiving posaconazole intravenously achieved significantly higher mean maximum and average concentrations of 4.13 mg/dL and 2.78 mg/dL, respectively, compared to those treated orally: 1.27 mg/dL and 1.09 mg/dL. The area under the concentration–time curve was also significantly higher in patients on IV solution, demonstrating a markedly better overall drug exposure. Interestingly, no effect on the time required to reach the maximum plasma concentration (T_max_) was observed (*p*-value = 0.428).

Patients administered ondansetron or levetiracetam presented with significantly lower C_max_, C_avg_ and AUC_tf_ values than those not using these medications. However, despite obtaining a *p*-value indicating a statistically significant correlation, drawing satisfactory conclusions was difficult due to the small number of patients receiving these drugs concomitantly with posaconazole in the study group.

Patients with hypoalbuminemia achieved slightly lower C_max_ and C_avg_ values, as well as a lower AUC_tf_, but the highest maximum PPC was also observed in patients with hypoalbuminemia. Among the 17 patients with hypoalbuminemia, 11 (65%) children achieved a PPC below the reference range. In two children, the levels of PPC were too high. Hypoalbuminemia was found to be a risk factor predisposing children to extreme values of PPC.

The analysis did not show a significant impact on the posaconazole PK parameters of the other examined variables, such as age, sex, the type of diagnosis, treatment involving HCT or CAR-T cell therapy, total parenteral nutrition (TPN), the occurrence of gastrointestinal problems (diarrhea, vomiting, mucositis) or other assessed drugs (e.g., ciclosporin A, proton pump inhibitors, corticosteroids).

In the second part of the study, the distribution of therapeutic and non-therapeutic concentrations in the study group was similar to that in the first part. In total, 51% of the patients (61/120) reached the target posaconazole plasma concentration (0.7–3.0 mg/L), but the percentage of children with an insufficient PPC was also significant, at 45% (54/120). Only 4% of patients (5/120) presented with a PPC above the upper limit. According to the results, in 49% (59/120) of patients, posaconazole dose modification was demanded. In 33 out of 59 patients, an attempt at dose modification could not be made due to the replacement of posaconazole with an alternative medication, the patient’s discharge or their movement to a different treatment center. Therefore, the modification of the dose was performed in only 26 patients. In 13 patients, adjustment of the dose by 30% was successful, resulting in the achievement of a PPC within the therapeutic range (Table 4). In most cases, only one modification was needed, while in two patients, a satisfying result was reached after two consecutive adjustments. In 12 patients, the escalation of the dose was needed, but a decrease in the dose was necessary in only one patient.

In 13 patients, the PPC remained constantly below 0.7 mg/L, despite the escalation of the dose or a change in the administration route from oral to intravenous (Table 5).

In a dozen patients, different posaconazole plasma concentration values were observed, despite no change in the dosage. None of the patients included in the trial were diagnosed with proven IFI during the observation period. IFI was suspected in three patients. In each of these cases, the PPC was within the reference range. In three other patients, antifungal treatment was escalated due to a fever of unknown origin that did not resolve after 72 h of broad-spectrum antibiotics administration. In two of these patients, the PPC was insufficient. In the study group, no adverse effects directly linked to the administration of posaconazole were reported. Among five children with a PPC exceeding the recommended range, three patients presented with elevated aminotransferases activity. The maximum values recorded were ALAT 128 U/L (normal range 0–45 U/L) and ASPAT 96 U/L (normal range 0–35 U/L).

In the statistical analysis, no correlation between PPC and the dose of the drug [mg/kg] was found (correlation of 0.636 in Kruskal–Wallis test). No significant relationship between the patient’s age (*p*-value = 0.115), sex (*p*-value = 0.437) or the type of diagnosis (*p*-value = 0.434) and the occurrence of a non-therapeutic posaconazole concentration in plasma was revealed.

Figure 1 illustrates the distribution of posaconazole concentrations across different age groups.

The significant impact of a few factors on the PPC was determined. Transplant patients, on average, achieved a lower PPC compared to patients treated without HCT involvement (0.84 mg/L vs. 1.27 mg/L). In the study, it was shown that an insufficient PPC is more common in patients undergoing HCT compared to those not undergoing a transplant procedure (57% vs. 34%). It was observed that the use of TPN, the presence of mucositis and other gastrointestinal pathologies were significantly more frequent in the transplant patient group. It appeared that transplant-related complications significantly influence the pharmacokinetics of posaconazole and the achievement of appropriate drug concentrations in plasma.

In the group of CAR-T patients, 6 out of the 11 children presented with a PPC below the reference range, with two patients showing extremely low levels (from 0.05 mg/L to below the detection threshold). However, the statistical analysis did not reveal a significant correlation between treatment involving CAR-T cell immunotherapy and the occurrence of non-therapeutic posaconazole concentrations in plasma (*p* = 0.72 in the chi-square test and 0.08 in the Mann–Whitney test).

The use of total parenteral nutrition was linked to lower posaconazole plasma concentrations. In 7 out of the 10 patients on parenteral nutrition receiving the posaconazole oral suspension, insufficient posaconazole levels were recorded. The analysis confirmed the statistically significant impact of TPN on the PPC, with a *p*-value of 0.011 in the Mann–Whitney test. However, no correlation was found in the chi-square test (*p*-value of 0.201).

When assessing the impact of the albumin concentration (3.5–3.0 g/L; >3.5 g/L; and <3.0 g/L) on the PPC, a significant correlation of 0.049 in the Kruskal–Wallis test was noted. The lowest average posaconazole plasma concentration was observed in patients with an albumin level in the range of 3.0–3.5 g/L. The highest PPC was noted in patients with albumin levels below 3.0 g/L. Similarly, as shown in part I, it was demonstrated that a deficiency of this carrier protein may result in both an insufficient and excessively high PPC.

Among the variables related to gastrointestinal pathologies, only diarrhea demonstrated a statistically significant impact on the PPC in the Mann–Whitney test (0.034). In the same test, mucositis (0.610) and vomiting (0.870) showed no association with the posaconazole concentrations.

A significant correlation between the type of diagnosis and the PPC was not shown. However, children with primary immunodeficiency syndrome (IDS) seem to be at a particularly high risk of an insufficient PPC, as seven out of eight patients presented with a concentration below the therapeutic range.

Certain drug-to-drug interactions for medicaments co-administered with posaconazole were analyzed. A relevant correlation was determined for methylprednisolone (*p* < 0.001), hydrocortisone (*p* < 0.001), proton pump inhibitors (*p* = 0.002), foscarnet (*p* = 0.039) and rifampicin (*p* = 0.005) in the chi-square test and/or Mann–Whitney U test. The co-administration of posaconazole with other medications had no impact on the PPC (e.g., ondansetron, levetiracetam, ciclosporin A). Proton pump inhibitors were found to be correlated with a lower PPC, as expected. Six of the thirteen patients in the first part of the study and eighteen of the twenty-six in the second part of the study administered IPP concomitantly with posaconazole oral suspension had an insufficient PPC. A significant correlation between the PPC and the use of proton pump inhibitors was observed in the second part of the study. In the logistic regression analysis, a 76% probability of non-therapeutic PPC was demonstrated.

A significant correlation between the intravenous administration of glucocorticoids and T_max_ was found in the statistical analysis of data from the first part of the study, with a *p*-value of 0.033 in the Mann–Whitney test. For other PK parameters, no significant correlation was observed. In the second part of the study, 32 patients received glucocorticoids. Statistical analysis demonstrated a significant correlation between the PPC and the administration of methylprednisolone and hydrocortisone. No significant impact on PPC was found for other glucocorticoids. Higher posaconazole levels were noted in patients receiving hydrocortisone, while the use of methylprednisolone was associated with a lower posaconazole concentration. All 10 patients receiving methylprednisolone due to GvHD had an insufficient PPC. While only three patients had symptomatic intestine involvement in GvHD, the remaining children presented with skin GvHD only. Hydrocortisone in concomitant therapy with posaconazole was related to a higher PPC, with the chance for a non-therapeutic PPC of 14% in the logistic regression analysis.

In the study, three patients co-administered rifampicin with posaconazole presented with an extremely low or unmeasurable PPC, which did not improve significantly despite the increase in the dosage. Statistical analysis confirmed a significant correlation between the PPC and the coadministration of rifampicin, although satisfactory conclusions could not be drawn due to the very limited number of patients. The impact of foscarnet administration on PPC could also not be assessed due to the limited number of patients (5), despite the statistically relevant correlation.

The available literature data and DDI databases do not indicate significant interactions between posaconazole and levetiracetam [12,13]. Levetiracetam was used in coadministration with posaconazole in 12 patients in both parts of the study, among which seven achieved an insufficient PPC ranging from 0.2 mg/L to below the detection threshold of the method, i.e., <0.05 mg/L. In the statistical analysis conducted for patients in the first part of the study, the significant impact of levetiracetam administration on the C_max_, C_avg_ and AUC_tf_ of posaconazole was observed, with *p*-values of 0.032 in the Mann–Whitney U test. No significant impact for the influence of levetiracetam on the posaconazole T_max_ was found. No significant effect of levetiracetam administration on the PPC in the second part of the study was confirmed.

Polymedication was not related to an increased chance of having a PPC out of the therapeutic range, but the number of medications with the potential for DDI with posaconazole displayed an influence on the T_max_ of posaconazole (*p*-value = 0.032). Factors with a significant correlation with the PPC are summarized in Table 6.

## 4. Discussion

The results of this study, as well as publications by other authors, highlight that insufficient posaconazole plasma concentrations (PPCs) are a prevalent problem among pediatric patients receiving antifungal prophylaxis with posaconazole oral suspension [15,16,17,18]. Posaconazole-treatment-related toxicity is rare [19,20,21]. No serious side effects were observed in this study. Nevertheless, the issue of adverse effects and determining the toxic dose of posaconazole requires further research.

Despite the significant intra-individual variability in posaconazole bioavailability and the high percentage of insufficient PPCs observed in children, posaconazole is considered an effective and safe medication for preventing invasive fungal infections (IFIs) in immunosuppressed pediatric patients. Several studies on the relationship between dosing, the PPC, and the clinical efficacy of posaconazole prophylaxis in pediatric patients have been published in recent years. Mathew et al. reported that, among a relatively small study group of 32 pediatric oncological patients, 50% had an insufficient PPC [16]. In the study by Arrieta et al., 43% of patients presented with subtherapeutic posaconazole levels, with a cutoff point set at <0.5 mg/L [15]. The results of Vicenzi et al. showed that 37% of 97 pediatric oncohematological patients achieved an insufficient PPC [18]. The study by Jia et al. showed a definitely higher percentage of children with therapeutic posaconazole levels (≥0.7 mg/L), reaching 74.4% among 78 patients [22]. Achieving an appropriate PPC is crucial in assuring effective antifungal prophylaxis. However, the target concentration in primary prophylaxis in children has not been defined yet. The range most often recommended is 0.7–3.0 mg/L, but some authors suggest that an even lower threshold of 0.5 mg/L could be applicable [11,22]. In the study by Dolton et al., 12 out of 72 patients receiving posaconazole prophylaxis developed a fungal infection. The posaconazole concentration was significantly lower in these patients compared to those without an infection (0.289 versus 0.485 mg/L) [23]. Jia et al. also observed a significant correlation between the development of IFI and the PPC in patients on prophylaxis; 9 out of the 75 studied patients developed confirmed or probable breakthrough invasive fungal infection. The PPC in these patients ranged from 0.13 to 0.70 mg/L, with a median of 0.43 mg/L, and was significantly lower than in patients without IFI (0.12–2.10 mg/L with a median of 1.2 mg/L) [22]. All breakthrough infections occurred in patients who had posaconazole plasma concentrations <0.5 mg/L. The absence of a demonstrated correlation between the subtherapeutic PPC and the increased incidence of IFI in this study is likely attributable to the small study group and the relatively short observation period. This study, along with other works, highlights the difficulty of achieving and maintaining a PPC at the target level throughout the treatment period. These challenges result from dynamically changing clinical circumstances, the significant variability in drug bioavailability and the difficulty of predicting the PPC in children [4,22]. In the study by Jia et al., it was shown that the variability in concentration among patients who had the highest number of measurements (five or more) was as much as 28.75% [22].

Optimizing posaconazole oral suspension prophylaxis in pediatric patients is not a simple task. In total, 50% of children benefited from dose modification in this study. According to Arrietta et al., neither an increase in dose nor in the frequency of drug administration was more effective in optimizing posaconazole exposure, especially in younger patients [15]. Adjusting the dosage by approximately 30% of the previously used dosage was chosen in this study, considering the possible causes that could lead to reduced drug exposure in pediatric hematooncological patients (e.g., decreased absorption due to the inflammatory conditions of the gastrointestinal mucosa, intestinal involvement in GvHD). Moreover, in pediatric patients, the dosing of posaconazole based on body weight usually does not exceed 800 mg per day, a dose at which the saturation of absorption effects has been described in adult patients [24,25]. The failure of intervention might be due to posaconazole’s involvement in drug–drug interactions, impaired gastrointestinal function or potential compliance issues.

Although in this study the correlation with the medication dose and PPC was not determined, other publications have reported that doses higher than currently recommended, exceeding 20 mg/kg, may influence the clinical efficacy of therapy and are required to ensure the target PPC. In this study, there was no difference in the primary dosages between patients with the target PPC and those with an insufficient PPC—16.26 mg/kg/day and 16.70 mg/kg/day, respectively. In patients with PPC > 3.0 mg/L, the initial dose was higher—17.75 mg/kg/day. In the study by Mathew et al., it was shown that the median initial dose of posaconazole in patients who achieved a concentration of ≥0.7 mg/L was 22.8 mg/kg/day, while it was significantly lower (15.8 mg/kg/day) in patients who did not reach the target threshold. It is important to note that only 15 out of 32 patients were receiving posaconazole for prophylaxis, while the remaining patients were receiving posaconazole for therapeutic indications. However, no statistically significant difference between the dose and PPC was demonstrated [16]. Similar results were obtained in the study by Bernardo et al., where patients with an insufficient PPC received lower doses of the drug than those who achieved the target concentration (12.9 mg/kg and 20 mg/kg, respectively). Posaconazole was dosed at 18–24 mg/kg/day for patients weighing < 34 kg and 800 mg/day for patients weighing > 34 kg in the IFI treatment regimen [26]. This may indicate the need to establish different initial posaconazole dosing schedules for different age groups. However, subsequent dosing should be based on TDM.

The difficulty in estimating the final posaconazole exposure and the challenge of predicting the consequences of modifying factors impact various aspects of treating IFI prophylaxis with posaconazole oral suspension. In this study, as well as in the publication by Jia et al., no statistically significant relationship was found between achieving the target posaconazole concentration and the patients’ age [22]. Similarly, no impact of gender on the PPC was demonstrated. Despite the fact that patients with ALL and AML are recognized to be particularly at risk of IFI, the analysis did not show a significant correlation between the type of diagnosis and PPC [13,27]. It was expected that hematopoietic cell recipients would demonstrate a higher rate of unsatisfactory PPC than patients treated without transplantation due to the intensive therapy and polymedication. This hypothesis was confirmed by the results of the second part of this study, where 57% of transplant patients achieved an insufficient PPC. Transplant patients reached, on average, lower PPCs than those treated without HSCT (0.84 mg/L vs. 1.27 mg/L). According to other publications, patients receiving posaconazole oral suspension who underwent HCT exhibited higher PPCs than patients treated for AML/MDS without transplantation (1.47 vs. 0.58 mg/L) [19,20]. This relationship was not observed with intravenous administration (1.56 vs. 1.47 mg/L) [28]. In the first part of this study, hematopoietic cell recipients achieved a PPC within the therapeutic range more frequently than patients not undergoing transplantation.

The statistical analysis confirmed the significant impact of parenteral nutrition on the PPC. The absorption of posaconazole oral suspension is best achieved when the drug is taken with a high-fat meal [29]. In patients orally administered posaconazole and fed entirely via parenteral nutrition, an insufficient PPC was frequently observed (80% and 70% of patients in the first and second parts of the study, respectively). As expected, due to the high percentage of posaconazole binding to proteins, the study confirmed a significant correlation between hypoalbuminemia and the risk of a non-therapeutic PPC. The study demonstrated that hypoalbuminemia significantly affects the pharmacokinetics and overall exposure to posaconazole. This issue is also highlighted by authors of recent studies conducted on adult patients receiving posaconazole in the delayed-release tablet formulation, presenting hypoalbuminemia as a risk factor for achieving subtherapeutic posaconazole plasma concentrations [30].

Similar to other studies, this research found that gastrointestinal disorders have a significant impact on the PPC. However, the statistical analysis found a significant impact on the PPC only in the case of diarrhea. The situation is particularly complex in patients with GvHD. During the study, it was noted that all patients receiving intravenous methylprednisolone achieved an insufficient PPC. Interestingly, the patients treated with this glucocorticoid, apart from three patients with intestinal GvHD, were treated due to the cutaneous form of the condition. A statistically significant relationship between the administration of methylprednisolone and the PPC was found, and a significant impact was noted also for T_max_. Patients receiving intravenous glucocorticoids required more time to reach the maximum plasma concentration of posaconazole. This prolonged time needed to reach the peak concentration, in conjunction with fast intestinal transit due to diarrhea and impaired absorption caused by mucositis, may result in lower exposure to posaconazole in the GvHD patient group, regardless of the organ in which GvHD manifests. Considering that patients with GvHD are at the highest risk of IFI, effective antifungal prophylaxis is particularly crucial in this group.

In the course of the analysis, clinically significant interactions were found with several medications concomitantly used with posaconazole. The significant impact of PPIs on the bioavailability of posaconazole was observed and confirmed previously. Despite the small patient group administered posaconazole and rifampicin, the very low PPC that they achieved reflects the clinical significance of the coadministration of both drugs. In countries such as Poland where BCG vaccination is mandatory and administered shortly after birth, tuberculosis prophylaxis in patients with immunodeficiency syndromes is very important. Therefore, rifampin-type medications may need to be used concurrently with posaconazole, with careful attention paid to close patient monitoring. Co-administration should be avoided. However, drug–drug interaction (DDI) databases indicate that in the case of co-administration with these two medications, ‘monitoring’ should be implemented, and the use of both drugs is not prohibited [14].

An important issue to explore is the use of levetiracetam in patients receiving posaconazole oral suspension. The available literature and drug interaction databases do not indicate identified interactions between these medications. However, during the project, attention was drawn to the frequent occurrence of unsatisfactory PPCs (58%) among patients administered levetiracetam concomitantly with posaconazole. A significant correlation was found between the coadministration of hydrocortisone with posaconazole and PPC. Patients receiving hydrocortisone achieved a higher PPC compared to those not receiving this medication. Nevertheless, despite the statistical significance, this finding does not seem to have substantial clinical implications, as excessively high concentrations across the entire study group appeared to be a marginal issue.

Creating a profile of a patient at the highest risk of subtherapeutic posaconazole concentrations is not straightforward. In the study, particular focus was placed on three groups of patients: patients with GvHD treated with methylprednisolone, IDS patients taking rifampicin, and children undergoing CAR-T cell therapy and receiving levetiracetam. Nevertheless, further studies on optimal antifungal prophylaxis involving larger patient groups and longer observation times during TDM are required to establish recommendations regarding the management of patients with unsatisfactory results for PPC measurements.

This study was affected by some limitations. This was a single-center study. The total number of patients included was high (171), but the subgroups characterized by specific variables were limited. Patients were selected by convenience, including children currently hospitalized in the clinic receiving posaconazole for at least 7 consecutive days. Significant variability in the clinical situation among the patients included in the study restricted the analysis of the studied parameters.

## 5. Conclusions

This research, as well as the studies published previously, highlight the need for therapeutic drug monitoring in posaconazole-based prophylactic regimens in pediatric patients. TDM improves the efficacy of the therapy as it enables the identification of patients in whom the implemented treatment may not be optimal. IFIs remain a challenge as they are still related to high mortality rates; thus, determining the impact on effective prophylaxis is of high importance. Issues related to the pharmacokinetics of posaconazole in children, the determination of optimal dosing regimens and the target PPC in pediatric patients require further studies. Multicenter, randomized trials are needed. Nevertheless, advances in the field contribute to newer posaconazole formulations, which offer more stable pharmacokinetic parameters such as delayed-release tablets. Unfortunately, they are still unavailable in Poland and, what is more, this formulation can also be non-optimal for the youngest children.

## Figures and Tables

**Figure 1 jof-11-00038-f001:**
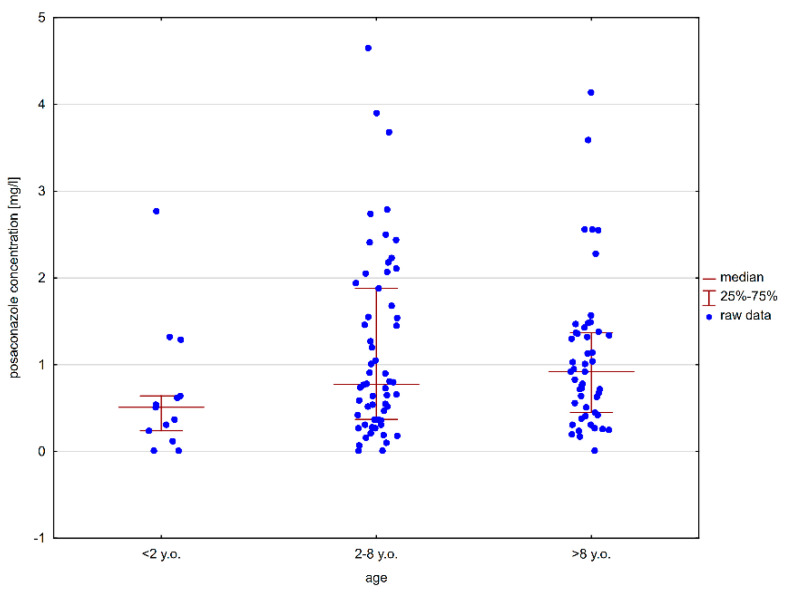
Distribution of posaconazole concentrations across different age groups.

**Table 1 jof-11-00038-t001:** Legend: yrs—years old, pts—patients, iv—intravenous, HCT—hematopoietic cells transplantation, CAR-T—chimeric antigen receptor T-cell therapy, ALL—acute lymphoblastic leukemia, AML—acute myeloblastic leukemia, MDS—myelodysplastic syndrome.

Patients’ Characteristics				
	Study part I—PK profiles		Study part II—TDM	
number of patients	51		120	
age	1–18 yrs, median 7 yrs		1–18 yrs, median 7 yrs	
sex	32 male (63%), 19 female (37%)		72 male (60%), 48 female (40%)	
diagnosis	malignant 40 pts (78%) -ALL 24 pts (60%) -AML 8 pts (20%) -other 8 pts (20%)	non-malignant 11 pts (22%)	malignant 91 pts (76%) -ALL 58 pts (64%) -AML 12 pts (13%) -MDS 8 pts (9%) -other 13 pts (14%)	non-malignant 29 pts (24%)
HCT	27 pts (53%) 8 pts GvHD		56 pts (47%) 9 pts GvHD	
CAR-T	3 pts (6%)		10 pts (8%)	
route of posaconazole administration	6 pts iv 45 pts oral		5 pts iv 115 pts oral	

**Table 2 jof-11-00038-t002:** Legend: maximum and average concentrations (C_max_, C_avg_) and area under the concentration–time curve (AUC_tf_), Min—minimal concentration, Max—maximum concentration, SD—standard deviation, PPC in [mg/L].

Selected Posaconazole PK Parameters in Correlation with the Route of Administration								
C_max_		Patients	Mean	Median	Min	Max	SD	*p*-Value
Route of administration	oral	45	1.27	1.11	0.09	6.07	1.15	0.002
	intravenous	6	4.13	4.13	1.44	6.99	2.14	
**C_avg_**		**Patients**	**Mean**	**Median**	**Min**	**Max**	**SD**	** *p* ** **-Value**
Route of administration	oral	45	1.09	0.97	0.06	4.56	0.96	0.006
	intravenous	6	2.78	2.65	1.13	4.48	1.38	
**AUC_tf_**		**Patients**	**Mean**	**Median**	**Min**	**Max**	**SD**	** *p* ** **-Value**
Route of administration	oral	45	8.76	7.80	0.50	36.45	7.66	0.006
	intravenous	6	22.27	21.18	9.02	35.87	11.02	

**Table 3 jof-11-00038-t003:** Legend: Pts no.—number of patients, Min—minimal concentration, Max—maximum concentration, SD—standard deviation, iv—intravenous, PPC in [mg/L].

Correlation Between Average Concentration (C_AVG_) of Posaconazole and Selected Factors								
C_avg_		Pts No.	Mean	Median	Min	Max	SD	*p*-Value
Route of admin.	oral	45	1.09	0.97	0.06	4.56	0.96	0.006
	iv	6	2.78	2.65	1.13	4.48	1.38	
Ondansetron	no	46	1.39	1.04	0.12	4.56	1.14	0.030
	yes	5	0.44	0.30	0.06	1.19	0.44	
Levetiracetam	no	48	1.36	1.04	0.17	4.56	1.13	0.032
	yes	3	0.33	0.12	0.06	0.80	0.41	
Hypoalbuminemia	no	34	1.38	1.11	0.18	3.81	0.94	0.029
	yes	17	1.12	0.57	0.06	4.56	1.44	

**Table 4 jof-11-00038-t004:** Legend: * optimal concentration was achieved in the second assessment after the escalation of the drug dose, ** the decrease in the concentration was achieved, nevertheless the PPC obtained was a little below the therapeutic range; LOD—level of detection.

Patients in Whom the Dose Modification Was Successful										
Patient	dose [mg]	C_0_ [mg/L]	dose1 [mg]	C_mod1_ [mg/L]	dose2 [mg]	C_mod2_ [mg/L]	dose3 [mg]	C_mod3_ [mg/L]	dose4 [mg]	C_mod4_ [mg/L]
4	2 × 120	0.62	2 × 160	0.87	2 × 200	0.58	2 × 200	1.32		
* 6	2 × 280	<0.05	2 × 360	0.15	2 × 360	1.41				
7	2 × 260	0.07	2 × 340	1.45						
8	2 × 160	0.21	2 × 200	0.29	2 × 260	1.48				
18	2 × 280	0.56	2 × 360	0.68	2 × 400	0.90	2 × 400	0.36	2 × 520	0.73
20	2 × 120	0.16	2 × 160	2.81						
23	2 × 300	0.38	2 × 360	0.85						
38	2 × 200	0.42	2 × 260	0.84	2 × 260	<LOD	2 × 320	0.21	2 × 360	1.40
57	2 × 200	0.66	2 × 240	0.71						
** 63	2 × 120	3.68	2 × 80	0.60						
66	2 × 300	0.31	2 × 360	1.09						
85	2 × 200	0.18	2 × 280	1.03						
95	2 × 160	0.47	2 × 200	1.10						

**Table 5 jof-11-00038-t005:** Legend: LOD—level of detection, once daily administration is for intravenous route.

Patients in Whom the PPC Remained Below the Therapeutic Range Despite the Escalation of the Dose or Change in the Route of Drug Administration From Oral to Intravenous										
Patient	dose [mg]	C_0_ [mg/L]	dose 1 [mg]	C_mod1_ [mg/L]	dose 2 [mg]	C_mod2_ [mg/L]	dose 3 [mg]	C_mod3_ [mg/L]	dose 4 [mg]	C_mod4_ [mg/L]
19	2 × 400	0.45	2 × 520	<0.10	2 × 300	0.12	2 × 400	0.45	2 × 400	0.37
25	2 × 80	0.37	2 × 120	<0.01						
26	2 × 80	<0.10	2 × 120	0.17	2 × 160	<0.10				
42	2 × 160	0.55	2 × 200	0.56						
48	2 × 280	0.68	2 × 360	0.37	2 × 400	0.38	2 × 400	0.26		
49	2 × 300	0.17	1 × 300 mg	0.69	1 × 300 mg	0.40				
51	2 × 80	0.51	2 × 120	0.21						
52	2 × 280	<LOD	2 × 280	<LOD	2 × 360	<LOD				
55	2 × 120	0.37	2 × 160	0.59	2 × 160	0.30	2 × 200	0.38		
83	2 × 200	0.28	2 × 280	0.49	2 × 360	<LOD				
91	2 × 300	0.41	2 × 400	0.10						
92	2 × 280	0.26	2 × 360	0.55						
105	1 × 150 mg	0.31	1 × 150 mg	<LOD						

**Table 6 jof-11-00038-t006:** Legend: IPP—proton pump inhibitor, HCT—hematopoietic cells transplantation, TPN—total parenteral nutrition.

Factors with Significant Correlation with PPC In Mann–Whitney Test								
		Number of Patients	Mean	Median	Min	Max	SD	*p*-Value
hydrocortisone	no	108	0.97	0.72	0.01	4.65	0.90	0.00057
	yes	12	1.95	2.00	0.59	3.90	0.96	
methylprednisolone	no	110	1.14	0.82	0.01	4.65	0.96	0.00083
	yes	10	0.33	0.29	0.01	0.68	0.21	
IPP	no	94	1.17	0.92	0.01	4.65	0.95	0.00194
	yes	26	0.70	0.39	0.01	3.90	0.86	
rifampicin	no	117	1.10	0.78	0.01	4.65	0.94	0.00499
	yes	3	0.05	0.01	0.01	0.12	0.06	
HCT	no	64	1.27	0.97	0.01	4.65	1.05	0.00840
	yes	56	0.84	0.61	0.01	3.59	0.75	
TPN	no	108	1.13	0.78	0.01	4.65	0.96	0.01132
	yes	12	0.53	0.28	0.01	1.54	0.54	
diarrhoea	no	113	1.10	0.78	0.01	4.65	0.96	0.03380
	yes	7	0.55	0.27	0.16	1.36	0.55	
albumin concentration	>3.5	75	1.02	0.78	0.01	3.90	0.84	0.04939
	3.0–3.5	36	0.92	0.68	0.07	2.74	0.79	
	<3.0	7	2.60	2.59	0.52	4.65	1.76	

## Data Availability

The data presented in this study will not be available due to privacy reasons.

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
