# Peer review of "Efficacy and Safety Assessment of Antifungal Prophylaxis with Posaconazole Using Therapeutic Drug Monitoring in Pediatric Patients with Oncohematological Disorders—A Single-Centre Study"

_jof, 2025, doi:10.3390/jof11010038_

Round 1

Reviewer 1 Report

The authors carried out a study to assess the efficacy and safety of posaconazole as antifungal prophylaxis in 171 pediatric oncohematological patients. First, they evaluated the pharmacokinetics of posaconazole. Then they estimated the posaconazole plasma concentration (PPC) and its impact on prophylaxis efficacy (prevention of IFI) and safety (posaconazole-related adverse events). The authors also investigated the potential correlations of a series of factors with PPC. For patients with subtherapeutic PPC, they modified posaconazole dosage when possible based on therapeutic drug monitoring (TDM), with a success rate of 50%.  I have the following comments:

11.       The authors divided the total of 171 patients into two groups, by including 51 patients in Study Part I for investigating posaconazole pharmacokinetics, and 120 patients in Study Part II for assessing the PPC and its impact on antifungal prophylaxis efficacy and safety.  I have a few questions about the study patient population: i) Were there any other inclusion/exclusion criteria for the authors to select patients for this study in addition to being pediatric patients with oncohematological disorders? Please clarify it and provide the related information briefly. ii) How were the 171 study patients selected & divided into the two study parts I & II (51 vs 120)? On what basis? iii) Could the authors provide a flowchart of the study participants? It would be very helpful for a better understanding of the design of this study.

2.       This appears to be a prospective study, but it is not clearly stated in the manuscript. Please clarify this point.

3.       Some basic information about posaconazole prophylaxis is missing. Could the authors provide the posaconazole prophylaxis durations for the patients in each study part, such as mean, median and range? For patients who received posaconazole oral suspension, is it possible to provide the daily dose information briefly, or a bit detail about Welzen’s recommendations (Line 83)?

4.       Regarding the statistical methods used in the study (Line 116-117), please clarify & differentiate the use of the following methods: chi-squared test, Mann-Whitney U test, and the Kruskal-Wallis test by indicating what kind of data were analyzed with each of these methods. For instance, categorical variables were compared using chi-squared test.

5.       While study Parts I & II each had their own specific study purpose, the data from both parts were also used to investigate the potential correlations between some relevant variables selected and the measured PPC. However, such analysis was performed on patients in study Parts I and II separately (Line 108-117). Statistically, it would be more appropriate to combine the data from Parts I & II and perform a single analysis for each variable on all the 171 patients. And one of the advantages of doing this is that the statistical power would definitely increase for the correlation analyses.

6.       Furthermore, to investigate the impact of the selected variables on PPC, the authors performed the correlation analysis between the selected variables and the measured PPC. Since the primary outcome of interest in the study is the PPC within the target range (0.7-3.0 mg/L), I would suggest grouping the 171 patients into 3 groups based on their measured PPCs at Co: PPC< 0.7 (subtherapeutic PPC) vs PPC within 0.7-3.0 (target PPC) vs  PPC > 3.0 mg/L, and focusing on comparing the selected variables between the first two groups: subtherapeutic PPC vs target PPC. Compared to Table 6, such comparisons might provide more relevant and direct insights into the correlations between the selected variables and having a target PPC.

7.       In Table 1 about patients’ characteristics, please add the percentage of each variable in addition to the patient numbers listed. For patients’ hematological malignancy, please add information about the type of hematological malignancy such as AML, ALL…, and HSCT and GVHD as they are basic information about this patient population.

8.       The Spearman correlation test performed in Study Part I showed significant correlations between the PPC at Co and those at other timepoints (C3, C5 and C8) (Line 131-132). Could the authors provide the related Spearman correlation coefficients considering they provide important information about the strength of these correlations?

9.       In the study, 13 patients who received the modification of the posaconazole prophylaxis based on TDM still had subtherapeutic PPC (<0.7 mg/L) afterward.  Did the authors investigate the possible reasons for this? For example, were these 13 patients more likely to be associated with factors that significantly contributed to the decrease in PPC?    

110.   The organization and writing of the manuscript could be improved.

1.       Line 91-92:  A PPC of 0.7-3.0 mg/L was considered the target range in the study. Please include the reference information.

2.       Line 162-163: Please clarify what data (mean or median?) were used to describe the Cmax and Cavg. It seems that the median was used to describe Cmax for patients receiving oral posaconazole, while the mean were used for other data descriptions. Please use the same measure for all of them consistently.

3.       Line 244: “Assessing the impact of albumin concentration (3.5 – 3.0 g/L; < 3.5 g/L; and < 3.0 g/L)”. “< 3.5 g/L” seems to be a typo and should be “> 3.5 g/L” instead. Please correct it.

Reviewer 2 Report

Dear authors, thank you very much for this interesting article.

Minor suggestions are in the pdf file.

Using italics in scientific names (fungal names) would be best.

Maybe you can describe in the results section in more detail those results associated with oral or IV administration instead of "first group" or "second group." 

kind regards

Detail comments are in the pdf file

Reviewer 3 Report

Liszka and colleagues present a manuscript regarding posaconazole prophylaxis among pediatric patients.

Comments:

-          In the Methods section, please state whether participants received a loading dose.

-          In clinical practice of adult patients, we often use a minimum posaconazole blood level of 0.7 as being therapeutic in a prophylaxis situation, and 1.0 as being therapeutic in a treatment situation. How do these values fit in with results from your study?

-          Did a retrospective look at initial posaconazole dosage by mg/kg lead to any later correlations with drug levels that were too high or too low? Do your data suggest that it would be helpful to initially dose at a certain mg/kg dose?

-          This reviewer would never co-administer a rifampin-type medication with an azole-type medication, as the azole would predictably have very low levels. How did the 3 patients in your study end up with co-administration of a rifampin-type medication? Wouldn’t that be malpractice?

-          At the end of the discussion, you report that patients were selected randomly (every 3rd patient, every 4th patient, or some other random sequence). Do you actually mean that they were selected by convenience, rather than by a random sequence?

-          The discussion is a little long and rambles a bit; this could be tightened up with some wordsmithing.

-          In the Abstract introduction, define the abbreviation TDM so that its use in the conclusions sentence will be recognized.

-          Some references are superscripted, others are not. Reference numbers are not offset by parentheses or brackets in the text: what is the journal style, and then this should be incorporated into the text.

-          In Table 1, please provide a footnote to state what the abbreviations mean (HSCT, CAR-T).

-          In Table 1, HSCT should be changed to HCT, as that is the preferred term.

-          In Table 1, give a little bit of information about what the study parts were examining. For example, where it says “Study part I”, you could list within that same cell “Study part I: PK profiles”. Where it says “Study part II”, you could list within that same cell “Study part I: TDM”.

-          In Table 3, should “Pts no” be “Pts no.”?

-          In Table 4, what is “patient PN”?

-          In Table 6, please provide a footnote to explain abbreviations.

-          The second paragraph of the discussion should be reviewed for proper referencing.

-          There ae numerous issues with extra spacing, use of commas, extra tabs, and extra carriage returns throughout the manuscript. This could be reviewed and corrected by a secretarial assistant with a good command of English language.

-          Reference 17 has an odd author listing that should be double-checked.

-          In the bibliography listing, if there is a doi, then the authors often omit page numbers from the print portion of the reference. Suggest adding those page numbers back in.

Reviewer 4 Report

This manuscript addresses a topic of significant relevance to managing fungal infections. The study is well-conducted, and the work is well-written and presented; therefore, it merits publication.

This manuscript addresses a topic of significant relevance to the management of fungal infections. The study is well-conducted, and the work is well-written and presented; therefore, it merits publication.

However, a few minor revisions are required:

The excessive use of acronyms, particularly in the abstract, hinders reader comprehension and impedes a clear understanding of the content. Please minimize acronym usage and ensure all acronyms are defined upon first use.

Fungal genera should be italicized throughout the manuscript. Please ensure adherence to standard taxonomic nomenclature.

Some references, such as those on page 2, lines 45, 51, and 52, are incorrectly formatted. Please review and correct the formatting of all references to ensure consistency with the target journal's style guidelines.

Round 2

Reviewer 3 Report

None. This is a fine paper.

The authors could add to the discussion that in their country, Poland, where BCG vaccination is mandatory and administered shortly after birth, tuberculosis prophylaxis in patients with immunodeficiency syndromes is very important so rifampin-type medications may need to be used concurrently. With careful attention to
close patient monitoring. Co-administration should be avoided. However, drug-drug interaction (DDI) databases indicate that in the case of co-administration of these two medications, "monitoring" should be implemented, and the use of both drugs is not prohibited (Sandherr & Maschmeyer, 2011).

- In Table 3, should “Pts no” be “Pts no.”? (The word "no" is actually an abbreviation for the word "number", so it should have a period at the end.

- In Table 5, what is “patient PN”?

Author Response

Thank you for all your comments and suggestions. The part about rifampicin in co-administration with posaconazole in Poland in the context of BCG vaccination was added to the discussion - that was good point to explain and develop this issue in the manuscript (marked pink). 

The corrections in the Tables were made (marked pink as well) - apologies that it was missed previously.

Kind regards,

Karolina Liszka